**Technology**

# Federated analysis of BRCA1 and BRCA2 variation in a Japanese cohort

## Graphical abstract

## Authors

James Casaletto, Michael Parsons,
Charles Markello, Yusuke Iwasaki,
Yukihide Momozawa,
Amanda B. Spurdle, Melissa Cline

## Correspondence

jcasalet@ucsc.edu

## In brief

Casaletto et al. developed containerized methods to analyze sensitive data without compromising the privacy of the original study participants. Co-occurrence of unclassified variants with known pathogenic ones provides evidence of being benign. This research serves as a proof-of-concept that is generalizable to other data types, file formats, and bioinformatic analyses.

## Highlights

- Federated methods enable scientific analysis of privacy-sensitive data

- Federated co-occurrence enabled novel BRCA genotype and phenotype assessment

- Our federated methods are generalizable to other genes, phenotypes, and file formats

- This approach can be applied to other sensitive data analyses such as family studies

 Casaletto et al., 2022, Cell Genomics 2, 100109
March 9, 2022 © 2022 The Authors.

CellPress

## Technology

# Federated analysis of BRCA1 and BRCA2 variation in a Japanese cohort

James Casaletto,[1,4,*] Michael Parsons,[2] Charles Markello,[1] Yusuke Iwasaki,[3] Yukihide Momozawa,[3] Amanda B. Spurdle,[2] and Melissa Cline[1]

[1]UC Santa Cruz Genomics Institute, Mail Stop: Genomics, University of California, 1156 High Street, Santa Cruz, CA 95064, USA
[2]QIMR Berghofer Medical Research Institute, 300 Herston Rd., Herston, QLD 4006, Australia
[3]Laboratory for Genotyping Development, RIKEN Center for Integrative Medical Sciences, 1-7-22 Suehiro-cho, Tsurumi-ku, Yokohama City, Kanagawa 230-0045, Japan
[4]Lead contact
*Correspondence: jcasalet@ucsc.edu

## SUMMARY

More than 40% of the germline variants in ClinVar today are variants of uncertain significance (VUSs). These variants remain unclassified in part because the patient-level data needed for their interpretation is siloed. Federated analysis can overcome this problem by "bringing the code to the data": analyzing the sensitive patient-level data computationally within its secure home institution and providing researchers with valuable insights from data that would not otherwise be accessible. We tested this principle with a federated analysis of breast cancer clinical data at RIKEN, derived from the BioBank Japan repository. We were able to analyze these data within RIKEN's secure computational framework without the need to transfer the data, gathering evidence for the interpretation of several variants. This exercise represents an approach to help realize the core charter of the Global Alliance for Genomics and Health (GA4GH): to responsibly share genomic data for the benefit of human health.

## INTRODUCTION

One obvious and well-studied example of how genetic variation can impact human health is the risk of cancer presented by pathogenic variation in the *BRCA1* and *BRCA2* genes. Pathogenic *BRCA1/2* variants greatly increase the risk of female breast and ovarian cancer (as reviewed)[1] and also confer significant risk of pancreatic, prostate, and male breast cancer (as reviewed).[1] Genetic testing that identifies a pathogenic variant in these genes enables individuals and their families to better understand their heritable cancer risk and to manage that risk through strategies such as increased screening, cascade testing of family members, and risk-reducing surgery and medication (as reviewed).[1] However, these risk-reducing strategies are not available to an individual found to carry a variant of uncertain significance (VUS), a rare variant for which there is insufficient evidence to assess its clinical significance. While individually rare, these VUSs are collectively abundant. As of May 2021, ClinVar,[2] the world's leading resource on the clinical significance of genetic variants, reports that 8,592/25,028 (34.3%) of *BRCA1/2* variants therein are designated as VUSs, while an additional 1,204 (4.8%) have conflicting interpretations. In other words, roughly 40% of *BRCA1/2* unique variants in ClinVar have no clear clinical interpretation. Meanwhile, there are many more variants that have been observed in individuals but are not yet in ClinVar: the Genome Aggregation Database (gnomAD)[3] includes an additional 35,635 *BRCA1/2* variants compiled from genomic

sequencing research cohorts. Patients of non-European ancestry are significantly more likely to receive a VUS test report from *BRCA1/2* testing,[4] a disparity that stems largely from historical biases in genetic studies.[5,6]

The VUS problem persists in large part because VUSs are rare variants; no single institution can readily gather a sufficient set of observations for robust variant classification. Data sharing would seem to be the natural solution, but it faces logistical challenges. Variant interpretation often requires some amount of case-derived information: clinical observations of the variant in patients and their families together with their cancer history. However, case-level data is sensitive and private and can rarely be shared directly because of regulatory, legal, and ethical safeguards.[7] Yet sharing data on rare genetic variants is critical for the advancement of precision medicine, as advocated by organizations including the Global Alliance for Genomics and Health (GA4GH),[8] the American College of Molecular Geneticists (ACMG),[9] and the Wellcome Trust.[10] Fortunately, most variant interpretation does not require the case-level data per se, but rather variant-level summaries of information derived from those data. The ACMG/AMP guidelines for variant interpretation,[11] which specify forms of evidence for interpreting genetic variants, indicate use of variant-level summary evidence including population frequencies (*BA1, BS1, PM2*), segregation of the variant and the disorder in patient families (*PP1, BS4*), case-control analysis (*PS4*), and observations of the VUS *in cis* and *in trans* with known pathogenic variants (*PM3* and *BP2*, depending on

the disorder). What is needed is an approach to derive this variant-level evidence from siloed case-level datasets without the need for direct access.

Federated analysis offers such an approach. Rather than an institution sharing its case-level data with external collaborators, those collaborators share an analysis workflow with the institution. The institution runs the workflow on their cohort, generating variant-level data that is less sensitive and can be shared more openly. This can yield valuable evidence for variant interpretation without the sensitive data leaving the home institution.[12] Container technologies support this approach by bundling the software and all its dependencies into a single module for straightforward installation and deployment on a collaborator's system.[13] These technologies include Docker,[14] Singularity,[15] and Jupyter.[16] Containers and workflows can be shared on the Dockstore platform[17] so that multiple institutions can execute the same software, promoting reproducibility.

We developed analysis workflows to mine tumor pathology, allele frequency, and variant co-occurrence data for *BRCA1* and *BRCA2* from breast cancer patient cohorts at RIKEN, derived from BioBank Japan.[18,19] This analysis allowed the assessment of new variant interpretation knowledge from a cohort that would not otherwise be accessible. In addition to generating new knowledge on these genetic variants, this yielded new knowledge on the genetics of the Japanese population, which is underrepresented in most genetic knowledge bases. Moreover, we have generalized our container approach to work with any genotype-phenotype combination of data.

## DESIGN

In principle, one could share access to a protected genomics dataset by transferring that data to a trusted third party, such as a secure cloud, but a dataset that contains personally identifiable information generally cannot or should not be moved from its secure source location. Indeed, the BioBank Japan data is prohibited from anonymous export. Federated analysis leaves the data securely in place and instead moves the analytic software (which tends to be many orders of magnitude smaller in size than a research cohort) to the data host institution. We designed our federated analysis software to be transparent, modular, and extensible. The analysis software creates multiple reports that capture data quality, associated phenotype, allele frequency, and variant co-occurrence.

Any researcher analyzing a dataset must first ensure that the data values are interpreted correctly; this is especially true when the researcher cannot interact with the data locally. The first report is the data quality report, which addresses that need by providing basic statistics (such as minimum, maximum, mean, mode, and median) and reporting any missing or unexpected data values. For this report, we provide a Javascript object notation (JSON) configuration file that defines each of the fields of interest, as exemplified here for the content of the tumor pathology file. The report could be used to check data quality for any delimited file, with or without a header. This data quality report represents a general solution that can be reused for other datasets. Document S1 includes two full examples of a data quality report.

The second report we generate is the genotype-phenotype report. This report is optional and can only be run when there exists both the variant call format (VCF) file as well as a phenotype tab-separated values (TSV) file. The purpose of this report is to associate a sample's genotype and phenotype directly in the same record. Document S1 includes two full examples of a genotype-phenotype report.

The third and last report is the variant frequency and co-occurrence report. It was written to summarize the variant counts stratified by patient group (affected versus control) for estimating allele frequencies and to report on VUSs that co-occur *in trans* either with known pathogenic variants in complex heterozygous genotypes or with themselves as homozygous genotypes. The program takes as input a VCF file and outputs JSON files with the variant counts and the co-occurring variant information. If associated phenotype data are provided, then our software will intersect those phenotype data with the genotype data in the VUS reports. This requires using a tab-separated file with the string "ID" as the primary key of this table whose values match those in the VCF file. Document S1 includes three full examples of variant frequency and co-occurrence reports.

To extend on the reporting functionality and generalizability, we provide the ability to integrate and call a custom, domain-specific report that can be leveraged to identify data anomalies in a known domain. This report is optional. In our research, we leveraged this feature to implement a tumor pathology report in which we calculate the number and proportion of triple-negative breast cancers of all breast cancers for which receptor status test results are available. This pathology report reads a tab-delimited file that is indexed by the sample identifier. Even though these sample identifiers are anonymized, we did not want to risk exposing any identifier in the results. Our tumor pathology report takes as input that same tumor pathology file and for each pathology feature outputs a summary of the number and proportion of patients stratified by pathogenic variant status, with an odds ratio, confidence interval, and Fisher's exact p value for the comparison. Additionally, the report includes a comparison of mean age at diagnosis (and entry) for the different patient groups. This can be extended to measure the statistics for any stratification of gene and pathology data. Importantly, this optional custom report can be independently used to validate that the researcher and the collaborator are reading and interpreting the data equivalently. In federated computing, the researcher never has direct access to the data, so any anomalies in the data could be identified if the researcher and collaborating institution agree to independently generate the same report and then compare the results. Indeed, we used this pathology report to validate our federated approach and to verify that there were no data anomalies that would preclude our analysis.

While our research focuses on VUSs in *BRCA1* and *BRCA2* genes and associated tumor pathologies, the software was written to work with any genotype-phenotype combinations of data. In Document S1, we provide an illustration of how one might assess genetic variation in cardiomyopathy by evaluating VUSs in the *MYH7* gene along with associated cardiac phenotype data. All the configuration is passed as command-line options to the program to define such parameters as gene name, whether the data are phased, and which human genome version

to use as genomic coordinates. Moreover, all the Python libraries required to run this code are included in the Docker container.

## Methods

### The dataset

Our analysis revolved around case-control association study data of individuals of Japanese ancestry.[18,19] These data reside at RIKEN and cannot be accessed outside of that institution. The dataset reports the variants in coding regions of 11 genes associated with hereditary breast, ovarian, and pancreatic cancer syndrome, including *BRCA1* and *BRCA2*. Additionally, the dataset reports the tumor pathology of the breast cancer patients, including estrogen receptor (ER), progesterone receptor (PR), and human epidermal growth factor receptor 2 (HER2) status. The controls within this cohort are individuals who were at least 60 years old when sequenced and who have neither personal nor family history of cancer. The variant data were stored in a VCF file and the associated phenotype (pathology) data were stored in a tab-delimited file. No other files were required for this analysis.

### Variant interpretation evidence

We developed Docker containers to collect data for two forms of evidence (ACMG code/codes designated in parentheses): allele frequencies (*BA1, BS1*) and variant co-occurrences (*BS2*). In addition, we estimated *in silico* predictions of variant pathogenicity (*BP4, PP3*) using the BayesDel method for annotation of predicted missense substitutions and insertion-deletion changes.[20]

### Allele frequencies

By the ACMG/AMP standards, the frequency of a variant in a large, outbred population can offer three different forms of evidence for variant interpretation. First, when the variant is observed at a far greater frequency than expected for the disorder in question, this is such a strong indicator of benign impact (*BA1*) that the variant can be considered benign without any further evidence. Second, when the variant's frequency does not meet the *BA1* threshold but is still greater than expected for the disorder, the frequency represents strong evidence (*BS1*) that can contribute to a benign interpretation. Third, when the variant is absent from controls or reference population datasets, its absence represents moderate evidence (*PM2*) that can contribute to a pathogenic interpretation.[11] While gnomAD is commonly used as a source of population frequencies, gnomAD 3.1 contains data from only 2,604 East Asian genomes,[3] while gnomAD 2.1 contains data from 9,977 exomes.[21] Similarly, gnomAD 2.1 contained 76 Japanese exomes, while the number of Japanese genomes in gnomAD 3.1 is unknown. Therefore, a Japanese biobank with tens of thousands of samples might plausibly contain additional evidence not available through gnomAD. When considering population frequencies, one must consider the source of the samples and whether individuals affected by the disorder are likely to be present in the dataset.[22] Accordingly, we evaluated the non-cancer subset of gnomAD and the control samples from BioBank Japan. Each ClinGen variant curation expert panel (VCEP) determines the precise rules for applying the ACMG/AMP standard to the genes and diseases under their purview, including the population frequency thresholds for *BA1* and *BS1* evidence. By the proposed rules of the *BRCA* ClinGen VCEP, the threshold for *BA1* evidence is an allele frequency of greater than 0.001, while the *BS1* frequency threshold is 0.0001 (A. Spurdle, M. Parsons, personal communication, March 12, 2021).

### In silico prediction

By ACMG/AMP standards, if multiple lines of computational evidence predict that a variant will impact either protein function or RNA splicing, that observation can contribute to a pathogenic interpretation (*PP3*). Conversely, if multiple lines of computation evidence predict that the variant will have no functional impact, that observation can contribute to a benign interpretation (*BP2*). We estimated the probability that the variant would impact protein function with BayesDel,[20] a meta-predictor that has been shown to outperform most others.[23] By the proposed rules of the *BRCA* ClinGen VCEP, a BayesDel score of less than 0.3 predicts a benign interpretation, while a BayesDel score of greater than 0.3 predicts a pathogenic interpretation.[24]

### In trans co-occurrence

In fully penetrant diseases with dominant patterns of inheritance, if one observes a VUS *in trans* (on the opposite copy of the gene) with a known pathogenic variant in the same gene in an individual without the disease phenotype, that observation represents evidence of a benign impact. For *BRCA2* (and more recently *BRCA1*), co-occurrences of two pathogenic variants in the same gene are associated with Fanconi anemia, a rare debilitating disorder characterized by deficient homologous DNA repair activity, bone marrow failure, early cancer onset, and a life expectancy that rarely extends past 40.[25] Consequently, when an older individual is observed with a *BRCA1* or *BRCA2* VUS as either a homozygous genotype or a compound heterozygous genotype (*in trans* with a pathogenic variant in the same gene), that observation suggests a benign interpretation for the VUS. One caveat is that most clinical sequencing does not report phase; any single co-occurrence of two variants might be *in trans* or *in cis*. However, if a VUS co-occurs with two different pathogenic variants in two different patients, one can assume that at least one of those co-occurrences is *in trans*.[26] Based on these clinical observations, VUS homozygosity or compound heterozygosity with a known pathogenic variant in an individual known or inferred to be without Fanconi anemia features provides strong evidence against pathogenicity (*BS2*).[23,25]

### Collaboration details

In advance of developing the containers, the authors communicated to determine which data were available and in which format the data were stored. In our research, the variant data were stored in a single VCF file with anonymized sample identifiers, and the pathology data were stored in a single TSV file indexed by the same sample identifiers. The data were already prepared in these files in the research that generated the data in the first place,[18,27] so no additional data preparation steps were required. RIKEN provided a pair of files (one VCF file and one tumor pathology TSV file) with bogus data to preserve privacy but simultaneously allow the University of California Santa Cruz (UCSC) researchers to develop their containers. As previously mentioned, the UCSC team initially developed the container to generate a tumor pathology report. When the UCSC team finished preparing the container for that report, they notified the team at RIKEN to download the container code and run it against the dataset. The instructions for running

the container are straightforward and well documented in the software repository. After a few iterations and email communications, the reports generated by each team were found to match exactly, thereby validating that accurate analysis could be performed on this data using a federated approach. Subsequently, the UCSC team developed the container to create the co-occurrence and allele frequency report along with the intersection and data quality report. Once those reports were generated, they were sent to the Queensland Institute of Medical Research (QIMR) team to analyze for variant interpretation. In all, the total amount of interaction required to collaborate was minimal, in part because the QIMR team had previously collaborated with the RIKEN team using this same data.[18]

### Analysis approach

We created our Docker containers with Python 3.73 code, which (1) collects observational statistics on tumor pathology, (2) gathers variant counts for estimating allele frequencies, and (3) identifies VUSs that either co-occur with a known pathogenic variant in the same gene or co-occur with themselves (i.e., homozygous VUSs). When reporting co-occurrences, we also reported the age of the patient to review data against expectations of age at presentation of Fanconi anemia. To identify VUSs, we checked the classifications provided by ClinVar and validated against the ClinGen-approved evidence-based network for the interpretation of germline mutant alleles (ENIGMA) expert panel in BRCA Exchange.[28] If the clinical significance was "Unknown," or if the variant did not appear in BRCA Exchange, then we labeled the variant a VUS. We applied this container to the BioBank Japan samples. We identified *BRCA1* or *BRCA2* variants that appeared as homozygotes and/or co-occurred with a known pathogenic variant in the same gene. Sequencing data were not phased, but details on the co-occurring variant(s) were provided to aid inference of whether a VUS was *in cis* or *in trans*.

## RESULTS

We describe here an example of how federated analysis can add information of value for variant interpretation. We analyzed a case-control study of Japanese individuals whose case-level data reside at RIKEN.[18,27] Because these data are not accessible to external researchers, the UCSC team developed analysis software, in the form of a Docker container, and shared it with the RIKEN team. The RIKEN team applied the container to analyze this cohort *in situ*, within their secure institutional environment, generating variant-level summary data that contain no personal information and can be shared more openly. The QIMR Berghofer team then applied these data to variant interpretation.

As an initial quality control exercise, we replicated the contents of Table S4 from a previous publication on these data[18] using the tumor pathology data. This table contrasts the patients with or without pathogenic variants in terms of factors, including family history of seven types of cancer; estrogen, progesterone, and herceptin receptor status; and age at diagnosis. We were able to replicate this table precisely, indicating that we were able to process the data accurately. This exercise also demonstrated that our container can be used to generate scientifically meaningful results. While this step was not mandatory for our analysis, we recommend it for the reasons just stated.

Subsequently, we applied the Docker container to analyze the complete patient cohort. We observed 19 *BRCA* variants that have not yet been interpreted by the ClinGen *BRCA1/2* expert panel. For each VUS, we reported its allele frequency in the controls and any observations of the VUS co-occurring with a known pathogenic variant in the same gene (Table 1). We also annotated variants for single-submitter curations in ClinVar.

Eleven VUSs met the standard for stand-alone evidence of benign impact (BA1) on the basis of the allele frequencies in the BioBank Japan controls; all of these VUSs were predicted bioinformatically to have benign impact (BP4). All 11 VUSs will meet the standard of benign interpretation on the basis of their frequency evidence from the Japanese cohort. Additionally, two of these variants (*BRCA1* c.4729T>C; *BRCA2* c.964A>C) were observed to co-occur with at least two different pathogenic variants in the same gene, evidence sufficient to apply the BS2 criterion. Of these 11 VUSs, four have single-submitter classifications in ClinVar as Benign or Likely Benign, five have conflicting interpretations, and two are designated by ClinVar as VUSs. Based on observations currently in gnomAD,[3] seven of these variants would have met the BA1 criterion, three would have met the BS1 criterion, and one was absent (meeting the PM2 criterion). For each of the variants present in gnomAD, East Asian was the continental population with the greatest allele frequency at the 95% confidence level (popmax),[29] a fact that itself adds confidence to the BioBank Japan observations. While seven of the variants could have been interpreted as benign using data in gnomAD, the federated analysis supported the interpretation of four additional variants. This greater sensitivity in the BioBank Japan results reflects the greater cohort size: while gnomAD contains 2,604 East Asian genomes and 9,977 East Asian exomes, the BioBank Japan control group contains 23,731 Japanese individuals.

Five VUSs showed strong evidence of benign impact (BS1) based on their BioBank Japan allele frequencies and evidence predictive of benign impact according to BayesDel (BP4). These five VUSs meet the standard of likely benign interpretation based on their frequency and bioinformatic evidence combined. Additionally, two of these VUSs had a single co-occurrence with a pathogenic variant in control individuals; while one should not put too much weight on any single homozygous observation, together with the BS1 and BP4 evidence, the data present a consistent picture of benign interpretation supported by multiple lines of evidence. One of these five variants is classified in ClinVar as likely benign, while the other four are classified as VUSs. Four of these VUSs would reach the BS1 evidence standard based on their gnomAD population frequencies, while a fifth is absent from gnomAD. The BioBank Japan analysis supports reclassifying five variants, only four of which could be reclassified using data in gnomAD.

Finally, three additional variants were each observed in a single heterozygous co-occurrence and have BayesDel scores predictive of benign impact (BP4). With one co-occurrence observation apiece, we cannot predict whether the co-occurrence is *in trans* or *in cis*, so these observations are not themselves sufficient for evidence of benign impact. However, these co-occurrences could contribute to benign evidence when and if the same VUSs are observed to co-occur with other pathogenic variant(s) in another cohort. These VUSs are rare variants absent

**Table 1. Summary of the variant data**

| | BRCA2 | BRCA2 | BRCA2 | BRCA1 | BRCA2 | BRCA2 | BRCA2 |
|---|---|---|---|---|---|---|---|
| Gene | BRCA2 | BRCA2 | BRCA2 | BRCA1 | BRCA2 | BRCA2 | BRCA2 |
| Variant (cDNA HGVS) | c.6325G>A | c.7052C>G | c.943T>A | c.4729T>C | c.4365A>G | c.6131G>T | c.964A>C |
| Variant (protein HGVS) | p.A2351G | p.A2351G | p.C315S | p.S1577P | p.A2351G | p.G2044V | p.K322Q |
| ClinVar classification (May 1, 2021) | B/LB | B/LB | B/LB | B/LB | LB | Conflict | Conflict |
| gnomAD 2.1.1 exome frequency (EAS) | 2.55E−03 | 1.87E−03 | 5.30E−03 | 2.65E−04 | Absent | 4.52E−04 | 4.31E−04 |
| gnomAD 3.1.1 genome frequency (EAS) | 2.39E−03 | 2.02E−03 | 5.03E−03 | 2.02E−04 | 2.01E−03 | 4.52E−03 | 2.41E−03 |
| ACMG/AMP code from gnomAD | BA1 | BA1 | BA1 | BS1 | BS1 | BA1 | BA1 |
| Biobank Japan frequency (Controls) | 1.46E−02 | 3.16E−03 | 1.56E−03 | 1.14E−02 | 4.64E−04 | 3.29E−02 | 2.31E−03 |
| ACMG/AMP frequency from BioBank Japan | BA1 | BA1 | BA1 | BA1 | BS1 | BA1 | BA1 |
| BayesDel score | −0.61 | −0.24 | −0.41 | 0.03 | −0.52 | −0.16 | −0.08 |
| Bioinformatic code | BP4 | BP4 | BP4 | BP4 | BP4 | BP4 | BP4 |
| ACMG/AMP class based on frequency and bioinformatics | B | B | B | B | LB | B | B |
| Gene | BRCA1 | BRCA1 | BRCA2 | BRCA2 | BRCA2 | BRCA2 | N/A |
| Variant (cDNA HGVS) | c.154C>T | c.811G>A | c.5969A>C | c.3395A>G | c.9733T>G | c.5660C>T | N/A |
| Variant (protein HGVS) | p.L52F | p.V271M | p.D1990A | p.K1132R | p.S3245A | p.T1887M | N/A |
| ClinVar classification (May 1, 2021) | Conflict | Conflict | Conflict | VUS | VUS | VUS | N/A |
| gnomAD 2.1.1 exome frequency (EAS) | 1.36E−03 | 1.32E−03 | 0 | Absent | Absent | 1.13E−04 | N/A |
| gnomAD 3.1.1 genome frequency (EAS) | 4.03E−04 | 1.21E−03 | 4.03E−04 | 0.000201 | Absent | Absent | N/A |
| ACMG/AMP code from gnomAD | BA1 | BA1 | BS1 | BS1 | PM2 | BS1 | N/A |
| Biobank Japan frequency (Controls) | 6.78E−03 | 6.28E−03 | 2.61E−03 | 3.75E−03 | 1.01E−03 | 1.69E−04 | N/A |
| ACMG/AMP frequency from BioBank Japan | BA1 | BA1 | BA1 | BA1 | BA1 | BS1 | N/A |
| BayesDel score | 0.14 | 0.06 | −0.08 | −0.2 | −0.47 | −0.29 | N/A |
| Bioinformatic code | BP4 | BP4 | BP4 | BP4 | BP4 | BP4 | N/A |
| ACMG/AMP class based on frequency and bioinformatics | B | B | B | B | B | LB | N/A |
| Gene | BRCA2 | BRCA2 | BRCA2 | BRCA2 | BRCA2 | BRCA2 | N/A |
| Variant (cDNA HGVS) | c.2672T>A | c.587G>T | c.8040C>G | c.358G>A | c.3983G>A | c.6637T>C | N/A |
| Variant (protein HGVS) | p.V891D | p.S196I | p.D2680E | p.V120M | p.S1328N | p.S2213P | N/A |
| ClinVar classification (May 1, 2021) | VUS | VUS | VUS | Absent | Conflict | Conflict | N/A |
| gnomAD 2.1.1 exome frequency (EAS) | Absent | 1.78E−04 | Absent | Absent | 0 | Absent | N/A |
| gnomAD 3.1.1 genome frequency (EAS) | Absent | Absent | 0.000202 | Absent | Absent | Absent | N/A |
| ACMG/AMP code from gnomAD | PM2 | BS1 | BS1 | PM2 | PM2 | PM2 | N/A |
| Biobank Japan frequency (Controls) | 9.69E−04 | 4.64E−04 | 9.69E−04 | 0 | 0 | 0 | N/A |
| ACMG/AMP frequency from BioBank Japan | BS1 | BS1 | BS1 | PM2 | PM2 | PM2 | N/A |
| BayesDel score | −0.05 | −0.22 | −0.05 | −0.48 | −0.57 | −0.06 | N/A |
| Bioinformatic code | BP4 | BP4 | BP4 | BP4 | BP4 | BP4 | N/A |
| ACMG/AMP class based on frequency and bioinformatics | LB | LB | LB | VUS | VUS | VUS | N/A |

The HGVS terms reflect the NM_007294.3 transcript for *BRCA1* and NM_000059.3 for *BRCA2*. Variants are designated as B (Benign), B/LB (Benign or Likely Benign), LB (Likely Benign), Conflict (Conflicting Interpretations), VUS (Uncertain Significance), or Absent (Not Found). All variants scored against the BayesDel *in silico* predictor with a score of less than 0.3, within the BP4 scoring range. Additionally, two variants were observed to co-occur with two more pathogenic variants in the same gene, indicating that at least one of these co-occurrences must be *in trans*, which meets the standards of BS2 evidence. In *BRCA1*, we observed co-occurrences of c.4729T>C with c.1518del and c.188T>A and in *BRCA2,* we observed co-occurrences of c.964A>C with c.6952C>T, c.5645C>A, and c.6244G>T. While these VUSs had sufficient evidence for classification on allele frequencies only, these co-occurrences add further support to benign classification. We further observed co-occurrences of *BRCA2* c.5660C>T with c.1261C>T and c.4365A>G with c.7480C>T, evidence that could support a benign classification if these variants are observed in co-occurrences with different path-ogenic variants in other patient cohorts.

from gnomAD and have either conflicting or VUS interpretations in ClinVar.

## DISCUSSION

With this demonstration of federated analysis, we analyzed a protected cohort that we would not have been able to access directly, and we gathered knowledge on Japanese genetics to further the interpretation of *BRCA1/2* variants. Of 19 variants currently tagged as VUSs by the ClinGen BRCA expert panel, 12 were VUSs or conflicting in ClinVar. The suggested interpretations based on bioinformatic and frequency analysis assign a Benign or Likely Benign classification for 16 variants and highlight the value of extending data capture to a subpopulation not yet well represented in gnomAD. We also demonstrated the federated collection of variant co-occurrences and age at presentation; these data together provided further evidence supporting the Benign and Likely Benign variant interpretations. This analysis would not be feasible with the existing population frequency resources. For example, gnomAD, the resource selected by ClinGen as its standard, does not yet have a large Japanese cohort and now shares variant co-occurrences but without the patient age information that is needed for ruling out Fanconi anemia under ENIGMA's variant interpretation rules. These samples had been analyzed previously by the RIKEN and ENIGMA teams,[18,27] a fact that explains why an analysis of nearly 30,000 samples revealed only 19 VUSs. This federated analysis allowed us to revisit these data with updated classification criteria, as well as collect new evidence on variant co-occurrences. Further, by developing a tumor pathology report, we provide proof of principle that federated analysis can be designed to capture other clinical features relevant for variant interpretation. These additional data types are generally provided only in summary-level data presentations from published cohorts, at best. Additionally, this method can be applied to any other phenotype-genotype relationship that could benefit from otherwise siloed datasets.

We have also demonstrated that there are international sequencing projects that contain valuable information that could be applied today to variant interpretation but are not yet represented in major population data repositories. This is illustrated by the number of Japanese samples analyzed in this study (7,104 cases plus 23,731 controls) versus the size of gnomAD's East Asian cohort (2,604 genomes plus 9,977 exomes). In principle, the gnomAD and the related population genomics resources will grow with time to comprehensively represent all global populations. In practice, because of the high cost of processing external sequence data, gnomAD mostly imports data from cohorts that were sequenced at the Broad, where sequencing data are processed to a common standard (H. Rehm, personal communication, October 4, 2021). For these reasons, capturing global genetic diversity can benefit from gathering evidence from international sources. Because traditional data sharing is blocked by barriers, including laws that prohibit exporting genomic sequences, federated analysis can advance data sharing by limiting the scope of data to be shared to the information most needed.

In this instance, the data sharing was simplified by the fact that the RIKEN team had already assembled a case-control dataset on breast cancer, and in doing so, had already reduced the com-

plex phenotypic data to a set of simplified terms. In a typical variant interpretation scenario, the situation is more involved. In genetic testing, the phenotypic data is often absent, or provided in unstructured text fields that must be curated manually prior to any analysis—traditional or federated. Where phenotypic data is available in a structured, electronic form, federated analysis can be viable. The cancer diagnosis (or lack thereof) can be represented through Human Phenotype Ontology (HPO) terms,[19] with Disease Ontology[30] terms representing the tumor pathology. For example, if the phenotype file had represented the disease phenotype with HPO terms rather than the simplified representation, one might distinguish between cases and controls in the genotype-phenotype report by recognizing breast cancer cases with the HPO term HP:0003002 (Breast Carcinoma), or potentially the less specific HPO term HP:0100013 (Neoplasm of the Breast). Similarly, if the phenotypic data were associated with cardiomyopathy, one could use the HPO term HP:0001639 to represent hypertrophic cardiomyopathy as a phenotype, or the more general HPO term HPO:0001639 to represent cardiomyopathy. Structured models for phenotypic and genomic data exchange, such as Phenopackets,[31] increase the opportunity for federated approaches by improving the data interoperability. With the growth in standards developed by the GA4GH and other organizations and increasing adoption of electronic data standards worldwide,[31] this federated analysis model can be generalized and extended into more areas within genomics. Emerging GA4GH technologies including Beacon V2, Matchmaker Exchange, and Data Connect can suggest the presence of samples of interest in remote, siloed cohorts, such as cases with rare monogenic disorders. This federated analysis approach complements such approaches by allowing further analysis of these samples while safeguarding patient privacy.

While gnomAD is a comprehensive source of allele frequency data in genomic research,[27] our federated solution does not, per se, require using it. Any database deemed more appropriate for a particular use case or cohort may be used as the source of allele frequencies if the data are formatted in a VCF sites file. Similarly, we used ClinVar as our source of ground truth for variant classification, and the ClinVar database may be substituted with another classification database if the data are formatted properly. These data formats are discussed in the supplemental information.

### Limitations of the study

Federated computing is being widely adopted, but it does present its own challenges in data privacy and system security. Docker containers are, to an extent, "black boxes." In order to ascertain whether the analysis is truly both secure and privacy preserving, an auditor would need to carefully inspect the Dockerfile definition of the container as well as all the software that runs in the container. We mitigated this risk by writing our reports to local text files that could be examined by the RIKEN team before being shared externally. Additionally, we published the software as open source so it may be directly inspected by collaborators. A second, related problem is that one cannot readily determine whether software might damage or compromise the security of the system on which it runs. One promising solution to this problem is certification. Within the emerging field of applications security testing, there are software platforms that can dynamically assess the

system accesses of the software under test. While the current platforms are commercial, there will likely be an open-source version in time. Eventually, this may become an element of the GA4GH Cloud Testbed, currently under development. This testbed infrastructure will initially serve as a platform for testing compliance with GA4GH standards and will extend to encompass performance benchmarking. In the future, this platform could potentially report activity that suggests a security risk, such as the details of outgoing network or disk traffic; and publishing these certification results could fit well within the framework of container libraries such as Dockstore. As an immediate solution to this problem, collaborating institutions should run such otherwise unsecured containers in a virtual machine sandbox environment that is completely isolated from their internal network.

Another limitation of our approach is that it requires getting data into the format that our software recognizes, namely tab-separated files and VCF files. In other words, the software is not agnostic of the file format. Moving forward, we will be able to generalize this approach by leveraging the data standards under development by the GA4GH, which will allow methods to compute over generalized data representation models rather than restricting their input to specific file formats. In particular, the standards of the GA4GH Cloud Workstream are already making it easier to leverage software methods across many different computing platforms. Further development will facilitate the streamlined execution of containerized workflows, the representation of phenotypic data, and the sharing of genetic knowledge.

## STAR★METHODS

Detailed methods are provided in the online version of this paper and include the following:

- KEY RESOURCES TABLE
- RESOURCE AVAILABILITY
  - Lead contact
  - Materials availability
  - Data and code availability
- METHOD DETAILS

### SUPPLEMENTAL INFORMATION

### ACKNOWLEDGMENTS

We gratefully thank Gunnar Rätsch for instigating this project and the members of the BRCA Challenge Evidence Gathering Group for discussion on the analytical design. J.C. is supported by NHGRI grant U54HG007990 and NHLBI grant U01HL137183.

A.B.S. and M.P. are supported by funding from the Australian National Health and Medical Research Council (APP177524). Y.M. is supported by AMED under grant number JP19kk0305010 (to Y.M.). M.C. is supported by NCI grant U01CA242954 and BioData Catalyst fellowship OT3 HL147154 from the NHLBI through UNC-CH 5118777.

### AUTHOR CONTRIBUTIONS

Y.I. and Y.M. performed the research that generated the variant and pathology data. M.C., A.B.S., and Y.M. planned the analysis of the data. The docker container was developed by J.C. with input from Y.M. and technical guidance from C.M. Y.I. and Y.M. executed the container. M.C., A.B.S., J.C., and M.P. analyzed the results and prepared the manuscript. All authors reviewed the final manuscript.

### DECLARATION OF INTERESTS

The authors declare no competing interests.

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

## Technology

 CellPress

## STAR★METHODS

### KEY RESOURCES TABLE

| REAGENT or RESOURCE | SOURCE | IDENTIFIER |
|---|---|---|
| **Deposited data** | | |
| Sequence and phenotype data | Japanese Genotype-Phenotype Archive | Japanese Genotype-Phenotype Archive: JGAS00000000140 |
| **Software and algorithms** | | |
| Co-occurrence GitHub repository | This manuscript | https://github.com/BRCAChallenge/federated-analysis |
| Co-occurrence Dockstore repository | This manuscript | https://dockstore.org/my-workflows/github.com/BRCAChallenge/federated-analysis/cooccurrence |
| Python 3.7.3 | Python Software Foundation | https://www.python.org |
| Scikit-allel 1.3.1 | Miles et al.[32] | https://scikit-allel.readthedocs.io/en/stable/ |
| Pandas 1.3.2 | Pandas development team[33] | https://pandas.pydata.org/ |
| Bcftools 1.10.2 | Danecek et al.[34] | https://github.com/samtools/bcftools |
| Pyensembl 1.8.5 | N/A | https://github.com/openvax/pyensembl |

### RESOURCE AVAILABILITY

#### Lead contact
Further information and requests for resources should be directed to and will be fulfilled by the lead contact for this study, James Casaletto (jcasalet@ucsc.edu).

#### Materials availability
There are no materials that were generated in this study.

#### Data and code availability
- This paper analyzes existing data from BioBank Japan. The accession number for the dataset is listed in the key resources table.
- All original code has been deposited at GitHub and Dockstore and is publicly available as of the date of publication. URLs are listed in the key resources table.
- Any additional information required to reanalyze the data reported in this paper is available from the lead contact upon request.

### METHOD DETAILS

To run our container, Docker must be installed in the runtime environment at the institution where the data are stored. We tested our container on Docker versions 18.03 and 19.03. The container also requires the appropriate ClinVar VCF file (for GRCh37 or GRCh38) which can be downloaded from their HTTP or FTP site (https://ftp.ncbi.nlm.nih.gov/pub/clinvar/). We used the bcftools command to reduce the size of this file to include only the genes of interest. Last, the container requires the gnomAD sites VCF file which can be downloaded from their HTTP site (https://gnomad.broadinstitute.org/downloads). Again, we used the bcftools command to reduce the size of this file to include only the genes of interest.

We created a variant co-occurrence and allele frequency report for BRCA1 and BRCA2, but our software has been generalized to find co-occurrences on other genes. Users can specify which version of the human genome (37 or 38), the chromosome and the gene on which to find VUS co-occurring in *trans* with themselves or with known pathogenic variants. The software runs on both phased and un-phased data, though inferring the genotype phase from un-phased data requires VCEP expertise.

To determine variant classification, users must provide a delimited file with the following fields: Clinical_significance and Genomic_Coordinate_hg37 (or Genomic_Coordinate_hg38). Genomic coordinates must have the form of this example variant: "chr13:g.32314514:C>T," where this represents the variant on chromosome 13, position 32314514 which changes a C nucleotide to a T nucleotide. If the Clinical_significance field is defined as "Pathogenic," "Likely pathogenic," "Likely_pathogenic," or "Pathogenic/Likely_pathogenic," then we interpret that variant as being pathogenic. Similarly, if the Clinical_significance field is

defined as "Benign," "Likely benign," "Likely_benign," or "Benign/Likely_benign," then we interpret that variant as being benign. We interpret any other value in the Clinical_significance field as being of uncertain significance.

To successfully mine co-occurrence data, our code performs the following steps.

1. Read VCF files

The genomic variants are defined in a VCF file which our application reads using the read_vcf() method of the Python scikit-allel package. We store the variants in a Python dictionary which contains the chromosome, position, reference allele, and alternate allele along with the genotype. The variant classifications are defined in a VCF file which our application reads using the read_csv() method of the Python pandas package. We store these classifications in a Python dictionary which contains 3 sets: one for pathogenic variants, one for benign variants, and one for VUS. Last, the allele frequencies are defined in a VCF sites file which our application reads using the read_csv() method of the Python pandas package. We store these allele frequencies in the same Python dictionary as the genomic variants.

2. Find variants per sample

Our application uses multi-threading in Python to parallelize the construction of 3 lists of variants per cohort sample: benign variants, pathogenic variants, and VUS. The classification of variants is determined using the ClinVar VCF file.

3. Intersect variants with phenotype data

The phenotype data are defined in a tab-delimited file and are read using the read_csv() method from the Python pandas package. The ID field of the phenotype file is used to match keys in the dictionary of variants per sample. In this way, any phenotypic data can then be associated directly with those variants and not with the samples themselves.

4. Find and annotate co-occurring VUS

Our application examines the dictionary of variants per sample and the associated genotypes to determine if those VUS co-occur in trans with themselves (homozygous) or with known pathogenic variants (heterozygous). We use the pyensembl Python package to annotate the variants with information such as whether it is exonic or intronic, and whether the variant falls within the known boundary of the gene of interest. Our application then generates the reports which contain the homozygous co-occurring VUS, VUS co-occurring with known pathogenic variants, any associated phenotype data per VUS, and the allele frequency data.

