## [Document S2. Transparent peer review records for Casaletto et al. · Cell Genomics]

Federated analysis of BRCA1 and BRCA2 variation in a Japanese cohort

James Casaletto¹, Michael Parsons², Charles Markello¹, Yusuke Iwasaki³, Yukihide Momozawa³, Amanda B. Spurdle², Melissa Cline¹

Summary

Initial submission: Received : June 1st 2021

Scientific editor: Orli Bahcall

First round of review: Number of reviewers: 2
Revision invited : September 12th 2021
Revision received : October 21st 2021

Second round of review: Number of reviewers: 2
Accepted : February 9th 2022

Data freely available:

Code freely available:

This transparent peer review record is not systematically proofread, type-set, or edited. Special characters, formatting, and equations may fail to render properly. Standard procedural text within the editor's letters has been deleted for the sake of brevity, but all official correspondence specific to the manuscript has been preserved.

Referee reports, first round of review

Reviewer #1

This is an interesting article by Casaletto and colleagues describing a new approach to data sharing in support of variant classification. Data sharing with a Japanese breast cancer case/control cohort is an important example because data from Asian individuals is woefully underrepresented in publicly available scientific databases despite the large contribution to the world population. The authors also highlight a critical problem in variant classification, which is that we often need some aspect of case-level data, particularly disease phenotypes, in order to complete classification of the variant per the ACMG/AMP classification guidelines. However, it is exactly this type of data which is more often restricted from public sharing often leading to variants being left in the large variant of uncertain significance category. Stated below are my concerns/suggestions about the paper.

Major:

1. The Results section gives us no idea of the amount of work (with regard to personnel, approvals, other costs) that was required by the RIKEN team to implement the Docker software and what device this software was placed on. This is my most serious concern. The article appears to be written very much from the perspective of the authors who are not based at RIKEN. For this approach to be practical, the authors need to be much clearer on what steps the RIKEN laboratory had to take, what type of personnel were needed, how much data cleaning was needed, how many levels of approval, to make this work in addition to my related comment below.
2. The main features of tumor pathology (actually ER/PR/HER2 status - unique to breast cancer) and co-occurrence (co-occurring with other BRCA1 or BRCA2 pathogenic variants) are specific to analysis of BRCA1/2 variant interpretation and not necessarily others. The authors need to provide more emphasis and potentially examples on how this structure could be modified to be used for other genes as the different ClinGen expert panels referred to in the article use different structures for variant classification. For example, co-occurrence can be useful for other genes but in practice is rarely implemented in as powerful a way as for BRCA1/2 interpretation. Perhaps selecting another gene (the authors are involved in several panels) and at least schematically showing how this would be done would be useful.
3. What was surprising to me was the lack of discussion of patient phenotypes. That is what is most often sought after to resolve rare variants. Although phenotypes are specific to the gene and disorder under question, HPO terms are generally used by expert panels to specify the phenotypes they are looking for, e.g. hearing loss versus, specific cancer diagnosis and age. Again, in order to generalize the use of this structure the authors might want to give some examples of how specific HPO terms could be transferred, even if the RIKEN dataset was all breast cancer patients and so the phenotype was predetermined.
4. The authors may want to comment that this type of data-sharing structure may work better for biobanks and biorepositories where the host institution has already put the combination of phenotype and genotype data together in a structured manner or perhaps for a diagnostic laboratory that requires highly structured data on requisition forms. Otherwise, a major problem with variant classification is that much of phenotype data is often not available to a testing laboratory, e.g., most people submitting samples for BRCA1/2 testing do not provide ER/PR/HER2 status to the testing laboratory.
5. The text and the Discussion seems to focus on the allele frequency data information. However, generally we want allele frequencies from controls and this data is less likely to be restricted. It appears that the utility of the Docker software structure is much more likely to be particularly useful for affected

cohorts, e.g. identifying alleles in patients with rare monogenic disorder the authors may want to consider this issue when revising the paper.

6. A clinical laboratory with their Laboratory information management system (LIMS) under CAP and CLIA inspection may be less willing to place an outside piece of software within their own system. This reluctance is becoming even greater with the substantial number of software systems that are being corrupted by malware encapsulated within other software modules. The authors should comment on this issue and what types of assurances were needed for RIKEN to take on this project.

Minor:

1. Awkward wording of sentence in the abstract. "We tested this principle with a federated analysis of breast cancer patients and controls from clinical data at RIKEN, derived from the BioBank Japan repository." You are analyzing the clinical data not the patients. Please rewrite.
2. Unusual referencing of the cancer risks for BRCA1/BRCA2 which is a huge literature; perhaps better to just cite one or two recent review.
3. The authors refer to gnomAD on multiple occasions appearing at times to be act as if it is the only other major database of allele frequencies. It would be useful to generalize these statements to include other resources such as UK Biobank, TopMed and other large projects that will be adding allele frequencies, particularly as other international biobank data becomes more available. Some of these resources, e.g., UK Biobank, also provide clinical data which is particularly useful and a significant problem when using gnomAD for more common adult onset diseases like breast cancer.

Reviewer #2

This is a well-written paper that reports on software that permits "federated analysis," that is, brings analysis tools to data rather than the reverse. Doing so may obviate consent and security challenges in some datasets since it avoids the export (download) of the data itself. The appeal of a federated approach is that it (theoretically, at least) will liberalize the availability of data that may otherwise be "locked up" by these (very reasonable!) constraints. This is especially appealing for increasing access to genetic variation in non-European populations that, to date, are the predominant type available in public databases (e.g., gnomAD).

In this paper the authors' own description works best to describe what was done: "We developed analysis workflows to mine tumor pathology, allele frequency, and variant cooccurrence data for BRCA1 and BRCA2 from breast cancer patient cohorts at RIKEN, derived from BioBank Japan (Momozawa et al. 2018). This analysis allowed the assessment of new variant interpretation knowledge from a cohort that would not otherwise be accessible. In addition to generating new knowledge on these genetic variants, this yielded new knowledge on the genetics of the Japanese population, which is underrepresented in most genetic knowledge bases."

I agree that this is an accurate description of their work.

The authors write clearly and the paper is well-organized. They discuss issues with the applying ACMG/AMP rules to variant interpretation, challenges with VUS, and granular knowledge on the difficulties of BRCA1/2 interpretation. I especially appreciated their comments on the potential disadvantages of federated analysis, which is increasingly promoted as a work-around to the very real issues of moving enormous datasets around.

To their credit, the authors do not hype or make special claims that this is "the first" of its kind. My

impression is that the apparent thoughtfulness of their approach will make this a useful model and tool for others.

Some issues to please comment on:

- 1) It is not clear to me if this software is a "one and done" case or could be adopted to other genes/data sources? How easily modifiable is the code so that the workflow could be applied to other genes, data sources, changes in ACMG/AMP variant interpretation rules?
- 2) It was easy enough to access the code on GitHub. Could some sample data files be provided to make the application easier for potential users?
- 3) The sentence on page 5-6 (of PDF available to me) is missing some words: "Accordingly, encoded our pathology report software in Structured Query Language (SQL) which is the most prolific language used in data analysis."
- 4) What is meant on page 5 about "hooks"?

Authors' response to the first round of review

Response to Reviewer #1 Comments:

Major:

1. The Results section gives us no idea of the amount of work (with regard to personnel, approvals, other costs) that was required by the RIKEN team to implement the Docker software and what device this software was placed on. This is my most serious concern. The article appears to be written very much from the perspective of the authors who are not based at RIKEN. For this approach to be practical, the authors need to be much clearer on what steps the RIKEN laboratory had to take, what type of personnel were needed, how much data cleaning was needed, how many levels of approval, to make this work in addition to my related comment below.

We added a subsection to Methods called "Collaboration Details" in which we describe how the three teams (UCSC, RIKEN, and QIMR) worked together to achieve our results. That text addresses the exact steps and effort needed by the RIKEN team.

2. The main features of tumor pathology (actually ER/PR/HER2 status - unique to breast cancer) and cooccurrence (co-occurring with other BRCA1 or BRCA2 pathogenic variants) are specific to analysis of BRCA1/2 variant interpretation and not necessarily others. The authors need to provide more emphasis and potentially examples on how this structure could be modified to be used for other genes as the different ClinGen expert panels referred to in the article use different structures for variant classification. For example, co-occurrence can be useful for other genes but in practice is rarely implemented in as powerful a way as for BRCA1/2 interpretation. Perhaps selecting another gene (the authors are involved in several panels) and at least schematically showing how this would be done would be useful.

We have provided three examples of usage of the method in the Supplemental Information section: one on BRCA1 without associated phenotype data, one on BRCA2 with associated phenotype (pathology) data, and one on MYH7 with associated cardiac phenotype data. We also added multiple references to those examples throughout the manuscript.

3. What was surprising to me was the lack of discussion of patient phenotypes. That is what is most often sought after to resolve rare variants. Although phenotypes are specific to the gene and disorder under question, HPO terms are generally used by expert panels to specify the phenotypes they are looking for, e.g. hearing loss versus, specific cancer diagnosis and age. Again, in order to generalize the use of this structure the authors might want to give some examples of how specific HPO terms could be

transferred, even if the RIKEN dataset was all breast cancer patients and so the phenotype was predetermined.

We thank the reviewer for this insightful suggestion, as the lack of discussion on patient phenotypes did in retrospect seem like a grave oversight. We have added a paragraph to the Discussion section on structured phenotype data, along with the applicable HPO terms. As mentioned above, in the Supplemental Information section, we now provide an example of how the current method might apply to a different gene and disorder: *MYH7* in cardiomyopathy.

4. The authors may want to comment that this type of data-sharing structure may work better for biobanks and biorepositories where the host institution has already put the combination of phenotype and genotype data together in a structured manner or perhaps for a diagnostic laboratory that requires highly structured data on requisition forms. Otherwise, a major problem with variant classification is that much of phenotype data is often not available to a testing laboratory, e.g., most people submitting samples for BRCA1/2 testing do not provide ER/PR/HER2 status to the testing laboratory.

Of course, the availability of phenotypic data is a requirement for this approach in particular and for variant classification in general. We have added a discussion on this to the Discussion section, noting that the phenotypic data provided in genetic testing varies greatly in its detail.

5. The text and the Discussion seems to focus on the allele frequency data information. However, generally we want allele frequencies from controls and this data is less likely to be restricted. It appears that the utility of the Docker software structure is much more likely to be particularly useful for affected cohorts, e.g. identifying alleles in patients with rare monogenic disorder the authors may want to consider this issue when revising the paper.

We understand this comment to say that we generally work with allele frequencies that are pre-computed over control populations, rather than indicating that the patient data from these control populations are less likely to be restricted. While there are control cohorts that are more open, such as 1000 Genomes, unrestricted patient data remains rare overall. If our understanding of this comment is correct, then we respond as follows. Yes, pre-computed allele frequencies from control populations are highly valuable but are not realistically all encompassing. As our manuscript illustrates, there can be valuable information in siloed biobanks which are not reflected in these control frequencies. Beyond that, yes, mining information on rare monogenic disorders is a compelling application of federated analysis, and we have added the following comment to the discussion section: "Emerging GA4GH technologies including Beacon V2, Matchmaker Exchange and Data Connect can suggest the presence of samples of interest in remote, siloed cohorts, such as cases with rare monogenic disorders. This federated analysis approach complements such approaches by allowing further analysis of these samples while safeguarding patient privacy".

6. A clinical laboratory with their Laboratory information management system (LIMS) under CAP and CLIA inspection may be less willing to place an outside piece of software within their own system. This reluctance is becoming even greater with the substantial number of software systems that are being corrupted by malware encapsulated within other software modules. The authors should comment on this issue and what types of assurances were needed for RIKEN to take on this project.

Great point! We added some guidance on this particular risk and how we mitigated it in the "Limitations of the Study" subsection of the Discussion section.

Minor:

1. Awkward wording of sentence in the abstract. "We tested this principle with a federated analysis of breast cancer patients and controls from clinical data at RIKEN, derived from the BioBank Japan repository." You are analyzing the clinical data not the patients. Please rewrite.

Thank you for spotting this. We updated the text accordingly.

2. Unusual referencing of the cancer risks for BRCA1/BRCA2 which is a huge literature; perhaps better to just cite one or two recent review.

We have “collapsed” the numerous references (1-7) into a single review reference (1).

3. The authors refer to gnomAD on multiple occasions appearing at times to be act as if it is the only other major database of allele frequencies. It would be useful to generalize these statements to include other resources such as UK Biobank, TopMed and other large projects that will be adding allele frequencies, particularly as other international biobank data becomes more available. Some of these resources, e.g., UK Biobank, also provide clinical data which is particularly useful and a significant problem when using gnomAD for more common adult onset diseases like breast cancer.

While we feel justified in our exclusive use of gnomAD for allele frequencies, our approach does not require using it, and another source of allele frequencies may suffice. We added this information to the Discussion section.

Response to Reviewer #2 Comments:

1) It is not clear to me if this software is a "one and done" case or could be adopted to other genes/data sources? How easily modifiable is the code so that the workflow could be applied to other genes, data sources, changes in ACMG/AMP variant interpretation rules?

We provide three working examples in the Supplemental Information section which show how to use our container: *BRCA* variants with and without phenotype data, and *MYH7* with phenotypic data (for cardiomyopathy). Collectively, these examples illustrate how this workflow can be applied to other genes and data sources. We acknowledge in the “Limitations of the Study” sub-section of the Discussion section that the file formatting requirements (VCF and TSV) can be a limiting factor in the usability of this solution. That said, getting data in and out of those formats is a straightforward task with the use of tools like BioConductor. All the variant classification was performed *ad hoc* and *per manum*: none of the ACMG/AMP rules are codified in the container software.

We have added a discussion to the Limitations section describing file formats and data types as a limitation of the current approach.

2) It was easy enough to access the code on GitHub. Could some sample data files be provided to make the application easier for potential users?

The examples that were included in the SI section are now also included in the code repository on GitHub and documented in the README file. Thank you for this suggestion.

3) The sentence on page 5-6 (of PDF available to me) is missing some words: "Accordingly, encoded our pathology report software in Structured Query Language (SQL) which is the most prolific language used in data analysis."

We removed the discussion on SQL, and so with that our miswording was also removed.

4) What is meant on page 5 about "hooks"?

“Hooks” in the context of software programming are ways that software developers allow for users of their software to integrate custom code (or “hook in custom code”) to the existing code. We take your point that this term is not known outside software developers, so we have updated the sentence accordingly.

Referee reports, second round of review

Reviewer #1

The authors substantially improved the manuscript in this revision and made it much more generalizable by showing several examples and providing more information about how the Riken site worked with the data.

Please correct one minor point. ClinGen does not limit the use of population data to only gnomAD. This sentence is incorrect. "and the only one used in the FDA-approved variant curation interface of ClinGen". This should be deleted. ClinGen has an emphasis on evaluating other population data, for example the PAGE program genotype frequencies are available in the ClinGen interface and there are published recommendations on what population data and size to use for PM2, BA1 and BS1 evidence codes. PMID 30311383

Reviewer #2

No further comments.

Authors' response to the second round of review

Reviewer #1:

The authors substantially improved the manuscript in this revision and made it much more generalizable by showing several examples and providing more information about how the Riken site worked with the data.

Please correct one minor point. ClinGen does not limit the use of population data to only gnomAD. This sentence is incorrect. "and the only one used in the FDA-approved variant curation interface of ClinGen". This should be deleted. ClinGen has an emphasis on evaluating other population data, for example the PAGE program genotype frequencies are available in the ClinGen interface and there are published recommendations on what population data and size to use for PM2, BA1 and BS1 evidence codes. PMID 30311383

We have updated that sentence as suggested:

While gnomAD is a commonly-used source of allele frequency data in genomic research²⁶, our federated solution does not, *per se*, require using it.